# Hemodynamic effects of high frequency oscillatory ventilation with volume guarantee in a piglet model of respiratory distress syndrome

Jagmeet Bhogal[1]*, Anne Lee Solevåg[2,3], Megan O'Reilly[1,4], Tze-Fun Lee[1,4], Chloe Joynt[1], Lisa K. Hornberger[1], Georg M. Schmölzer[1,4], Po-Yin Cheung[1,4,5]

1 Department of Pediatrics, University of Alberta, Edmonton, Alberta, Canada, 2 Department of Paediatric and Adolescent Medicine, Akershus University Hospital, Lørenskog, Norway, 3 Department of Paediatric and Adolescent Medicine, Oslo University Hospital, Nydalen, Norway, 4 Centre for the Studies of Asphyxia and Resuscitation, Royal Alexandra Hospital, Edmonton, Alberta, Canada, 5 Department of Pharmacology, University of Alberta, Edmonton, Alberta, Canada

* jagmeet.bhogal@ahs.ca

**Data Availability Statement:** All relevant data are within the manuscript and its Supporting Information files.

## Abstract

Respiratory failure is a common condition faced by critically ill neonates with respiratory distress syndrome (RDS). High frequency oscillatory ventilation (HFOV) is often used for neonates with refractory respiratory failure related to RDS. Volume guarantee (VG) mode has been added to some HFOV ventilators for providing consistent tidal volume. We sought to examine the impact of adding the VG mode during HFOV on systemic and cerebral hemodynamics, which has not been studied to date. A neonatal piglet model of moderate to severe RDS was induced by saline lavage. Piglets (full term, age 1–3 days, weight 1.5–2.4 kg) were randomized to have RDS induced and receive either HFOV or HFOV+VG (n = 8/group) or sham-operation (n = 6) without RDS. Cardiac function measured by a Millar® catheter placed in the left ventricle as well as systemic and carotid hemodynamic and oxygen tissue saturation parameters were collected over 240 min of ventilation. Mean airway pressure, alveolar-arterial oxygen difference and left ventricular cardiac index of piglets on HFOV vs. HFOV+VG were not significantly different during the experimental period. Right common carotid artery flow index by in-situ ultrasonic flow measurement and cerebral tissue oxygen saturation (near-infrared spectroscopy) significantly decreased in HFOV+VG at 240 min compared to HFOV (14 vs. 31 ml/kg/min, and 30% vs. 43%, respectively; p<0.05). There were no significant differences in lung, brain and heart tissue markers of oxidative stress, ischemia and inflammation. HFOV+VG compared to HFOV was associated with similar left ventricular function, however HFOV+VG had a negative effect on cerebral blood flow and oxygenation.

**Funding:** PYC received funding from the Canadian Institutes of Health Research, grant number MOP 130483 (http://www.cihr-irsc.gc.ca). The funder had no role in study design, data collection and analysis, decision to publish, or preparation of the manuscript.

**Competing interests:** The authors have declared that no competing interests exist.

## Introduction

Respiratory distress syndrome (RDS) is a frequent cause of respiratory failure in preterm infants [1]. European Consensus Guidelines on the Management of Respiratory Distress Syndrome suggest that mechanical ventilation optimizes lung volumes by providing an even distribution of tidal volumes to minimize atelectasis and hyperinflation. These goals can be achieved with either conventional mechanical ventilation (CMV) or high frequency oscillatory ventilation (HFOV) [2]. HFOV is mainly used as a 'rescue' treatment when CMV fails [3, 4]. HFOV enables ventilation by theoretical processes including the Pendelluft effect and longitudinal dispersion, in addition to the bulk convection shared with CMV, as described by Chang et al [5]. HFOV operates at a consistent mean airway pressure (MAP) compared to CMV. Given the potential for increased MAP to influence intrathoracic pressures, use of HFOV may result in reduced venous return, and, in so doing ultimately negatively impact cardiac output and oxygen delivery. This is a common clinical concern in the choice of HFOV over CMV [6–8]. Considering that many patients placed on HFOV are critically ill, understanding the full hemodynamic impact of such a ventilatory strategy is key [3].

Adding "volume guarantee" (VG) to CMV reduced lung damage in preterm infants and has become the standard of care [9]. With VG in CMV, the ventilator adjusts the pressures as needed to maintain a consistent volume, thereby adapting in real time to dynamic factors such as changing pulmonary compliance [10]. Over the past decade, while VG mode has been added to some commercially available HFOV ventilators, there has been limited research on the use of HFOV with VG (HFOV+VG), and these have largely represented pilot studies in a clinical setting focused on measuring stability of oxygenation and ventilation rather than hemodynamics [11–13]. There have been no prospective trials comparing short- and long-term outcomes, nor hemodynamic effects of HFOV and HFOV+VG. Given the influence of positive pressure ventilation on cardiac preload due to impedance of venous return, there may be related clinical consequences such as hypotension or end-organ hypoperfusion, leading to oliguria or lactic acidosis [6]. HFOV+VG titrates its amplitude pressures to maintain a set volume, and fluctuating amplitude on a constant background MAP may have additional deleterious effects on the cardiac preload.

Given the relative novelty of the HFOV+VG mode, in the present investigation we examined hemodynamic and pulmonary changes during ventilation by using a piglet model of RDS induced by saline lung lavage. We speculated that the varying pressures during HFOV+VG could affect left ventricular (LV) function and tested the hypothesis that HFOV+VG would decrease LV function compared to HFOV. We also compared the effects of HFOV and HFOV+VG on cerebral perfusion and tissue markers of injury in this neonatal piglet model.

## Methods

Full-term Yorkshire-Landrace piglets ranging from 1 to 3 days of age, weighing 1.5 to 2.4 kg, were used. The piglets were obtained from the Swine Research Technology Centre at the University of Alberta (Edmonton, AB, Canada). Ethics approval (AUP0000237) was obtained, and the study was conducted in accordance with the Canadian Council on Animal Care Guidelines and Policies with approval from the Animal Care and Use Committee for the University of Alberta. ARRIVE guidelines (Animal Research: Reporting of In Vivo Experiments) were followed in the design, conduction, and reporting of the experiments [14]. Piglets have been shown to have similar cerebrovascular, cardiovascular and respiratory systems to a human infant of 36–38 weeks' gestation. Although term piglets have higher pulmonary and systemic vascular resistances and a stiffer chest wall [15], the term piglet is an equivalent animal model to a late-preterm infant.

## Animal preparation

The piglets were positioned supine for the entire experiment. They were initially anesthetized with isoflurane and spontaneously breathing. A percutaneous oxygen saturation ($SpO_2$) sensor was placed on the right front limb to monitor $SpO_2$ throughout the experimental period. Two 5-French Argyle catheters (Sherwood Medical Co., St. Louis, Mo) were inserted into the femoral vein and artery to provide intravenous access, blood sampling and mean blood pressure monitoring, respectively. Piglets were then intubated via a tracheostomy and ventilated (Acutronic Fabian HFO; Hirzel, Switzerland) using CMV with default settings: positive end-expiratory pressure 6 $cmH_2O$, tidal volume of 10 mL/kg for goal end-tidal $CO_2$ partial pressure range 40–60 mmHg, ventilator rate 60/min, and inspiratory time 0.3 sec. $SpO_2$ was kept within 90–100%, glucose level and hydration was maintained with an intravenous infusion of 5% dextrose at 10 mL/kg/h. Anaesthesia was maintained with intravenous propofol 5–10 mg/kg/h and morphine 0.1 mg/kg/h. Additional doses of propofol (1–2 mg/kg) and morphine (0.05–0.1 mg/kg) were given as needed based on the neurobehavioral assessment of anaesthetic and analgesic level. The piglet's body temperature was maintained at 38.5–39.5°C using an overhead warmer and a heating pad.

A Millar® catheter (MPVS Ultra®, ADInstruments, Houston, TX) was inserted into the left ventricle via the left common carotid artery for continuous measurement of left ventricular pressure, composite, and segmental volumes, which were used for cardiac output calculation. The right common carotid artery was also exposed and encircled with a real-time ultrasonic flow probe (2mm; Transonic Systems Inc., Ithaca, NY) to measure cerebral blood flow. An INVOS near-infrared spectroscopy probe (Covidien, Minneapolis, MN) was placed on the forehead for monitoring cerebral oxygen saturation ($CrSO_2$).

Piglets were allowed to recover from surgical instrumentation until baseline hemodynamic measurements were stable (minimum of one hour). The sham-operated group (n = 6) underwent the same initial instrumentation and anesthesia as well as the subsequent 4-h monitoring period but did not undergo saline lung lavage. The timeline of the experimental protocol is shown in Fig 1.

## RDS induced by saline lung lavage

The lung lavage procedure was based on Lachmann's RDS model [16] with slight modifications, including lower volumes administered. Pilot experiments demonstrated worsening lung injury and decreasing oxygenation after the goal alveolar-arterial oxygen difference ($AaDO_2$) was achieved after the lavage was completed. The target $AaDO_2$ was between 300–450 mmHg to mimic moderate to severe RDS [17].

A Ringer's lactate 10 mL/kg bolus was given intravenously halfway through the stabilization period to improve tolerance of the saline lung lavage because bradycardia and hypoxia could

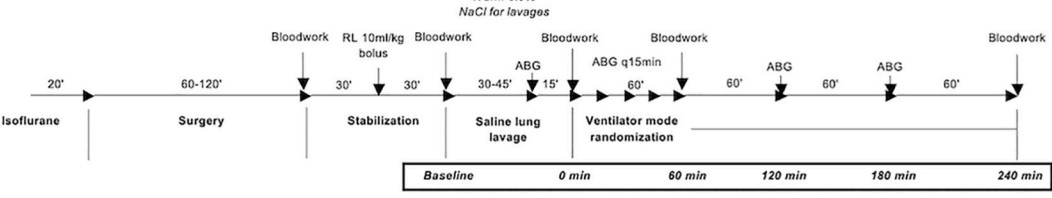

**Fig 1. Timeline of experimental protocol.** Baseline is at end of stabilization, and time 0 is at end of saline lung lavage with 60 min intervals afterwards ending at 240 min. Euthanasia and tissue harvesting occurred after 240 min collections. RL, Ringer's Lactate; ABG, arterial blood gas.

occur during the lavages. After stabilization, pre-warmed normal saline (approximately 39˚C) was injected through the endotracheal tube. The tube was then clamped and suctioned after 20 sec or sooner if the $SpO_2$ decreased to <40% or bradycardia of <90 bpm. The piglets remained on CMV between lavages. Lavage was repeated every 2 min until target $AaDO_2$ was achieved. The initial seven lavages were with 10 mL/kg to avoid an overshoot of the $AaDO_2$ target range; afterwards the volume was increased to 20 mL/kg until the $AaDO_2$ gradient target range was reached. Blood gases were assessed intermittently depending on the change in fraction of inspired oxygen ($FiO_2$) and $SpO_2$ with each lavage. Fifteen min after achieving the target $AaDO_2$, an arterial blood gas was done to confirm the target $AaDO_2$ gradient prior to randomization to HFOV or HFOV+VG.

## Administration of HFOV and HFOV with VG in piglets with RDS

Once at a stable target $AaDO_2$, the piglets were randomized to either HFOV or HFOV+VG (n = 8/group). A lung recruitment maneuver was not performed as it was not routinely used in our center and due to the potential destabilizing cardiopulmonary effects. HFOV with or without VG was immediately commenced with a mean airway pressure (MAP) of 2 $cmH_2O$ above the MAP that was required on CMV after the last lung lavage with stable target $AaDO_2$, inspiratory to expiratory ratio 1:2, frequency 10 Hz, and amplitude 30 $cmH_2O$. HFOV+VG was set with a tidal volume 2 mL/kg (maximum amplitude 45 $cmH_2O$). For piglets with hypercapnia (>60 mmHg), HFOV amplitude was changed by 5 $cmH_2O$ and tidal volume of HFOV+VG was adjusted by 0.5 mL/kg. For $SpO_2$ <90%, the $FiO_2$ was titrated preferentially over increasing the MAP. To avoid MAP being a confounding variable for hemodynamic effects, the MAP was only increased if the animal desaturated with $SpO_2$ <90% despite a $FiO_2$ of 1.0 with subsequent weaning of the MAP as rapidly as possible to a target MAP of 12 $cmH_2O$, while maintaining $SpO_2$ >90%.

Piglets were euthanized with an intravenous overdose of sodium pentobarbital (120mg/kg) after 240 min in the sham-operated and experimental groups.

## Biochemical assays

After euthanization, the left ventricle and right lower lung lobe were snap frozen in liquid nitrogen and stored in −80˚C until subsequent analysis. The brain was removed from the skull and placed in ice-cold 2-methylbutane for 10 min before being stored at −80˚C. A tissue core (100g) at the right parietal region was harvested for tissue biochemical assays. After extracting the tissues with either phosphate buffer (50mM containing 1mM EDTA, pH 7.0) for glutathione and cytokines or 6% perchloric acid and EDTA for lactic acid, protein concentration was quantified using the Bradford method. The concentrations of pro-inflammatory cytokines interleukin (IL)-8, and tumour necrosis factor (TNF)-α in brain tissue homogenates were determined using commercially available ELISA kits (P6000B, PTA00; R&D Systems, Minneapolis, USA). Cytokine concentrations were expressed relative to protein concentrations. The tissue levels of oxidized and total glutathione (GSSG and GSH) were measured using a commercially available assay kit (#703002, Cayman Chemical, Michigan). The lactate level was determined by a nicotinamide adenine dinucleotide enzyme-coupled colorimetric assay with spectrophotometry at 340 nm.

## Data collection and analysis

Hemodynamic parameters including heart rate, mean blood pressure, right common carotid artery flow and variables of LV systolic and diastolic functions that were measured by the Millar® catheter (stroke volume, cardiac output, ejection fraction, dP/dt max, dP/dt min, Tau,

end-diastolic volume and pressure) were continuously recorded using LabChart® programming software (ADInstruments, Houston, TX). Cardiac output was indexed for body weight as cardiac index (CI). Stroke volume and right common carotid artery blood flow were also indexed for body weight accordingly. Tau is a measure of diastolic function, whereas dP/dt min and dP/dt max refer to measures of minimum and maximum isovolumic contractility, respectively [18]. Arterial blood samples were taken at standardized time points throughout the experiment for monitoring blood gas changes and plasma lactate (Fig 1).

### Randomization

Piglets were randomly allocated to sham-operated, HFOV, or HFOV+VG groups. Allocation was randomized using a computer-generated randomization program (http://www.randomizer.org). Sequentially numbered, sealed, brown envelopes containing the allocation were opened during the experiments (Fig 1).

### Sample size calculation and statistical analyses

The primary outcome was cardiac function, as indicated by CI, at any time point in the study. Our hypothesis was that HFOV+VG would result in reduction in LV CI when compared to HFOV due to the fluctuation in amplitude and thus intrathoracic pressures to maintain a consistent tidal volume. Our power calculation involved finding a difference of 25% in CI between the experimental groups to be considered significant. Based on an α value of 0.05 and β error of 0.2, the sample size was calculated to be 8 piglets per group.

Data are presented as mean±standard deviation. Differences between groups were determined by two-way repeated measures ANOVA, which facilitated investigating interactions between ventilation modes and time points. Posthoc pairwise comparisons between modes and time points were done using the Tukey method. One-way ANOVA was used for the analysis of tissue biochemical markers. P-values are 2-sided and $p < 0.05$ was considered statistically significant. Statistical software used was SigmaPlot (version 11.0, Systat Software, San Jose, CA).

## Results

Twenty-two newborn mixed breed piglets were obtained on the day of the experiment and were randomly assigned to HFOV, HFOV+VG, or sham-operated groups. There were no significant differences in the baseline parameters between groups (Tables 1 and 2).

### Lung lavage and recovery

There were no significant differences in lavage time (53.8±6.9 vs. 61.1±18 min for HFOV and HFOV+VG; p = 0.31) or number of lavages (10.8±1.8 vs. 12.5±4.8 for HFOV and HFOV+VG; p = 0.35) between groups. Consequently, the $AaDO_2$ prior to the administration of HFOV or HFOV+VG was 402±39 and 423±79 mmHg for HFOV and HFOV+VG, respectively (p = 0.52). As shown in Fig 2, $AaDO_2$ of both HFOV and HFOV+VG groups remained significantly higher than their baseline values (240 min: 248±157 and 299±126 mmHg vs. 49±10 and 56±15 mmHg at baseline; respectively, both $p < 0.05$), and the $AaDO_2$ of sham-operated group at corresponding time points throughout the recovery period (240 min: 71±14 mmHg, $p < 0.05$) with no difference between the two treatment groups. The MAP followed a similar pattern with a higher MAP observed in both treatment groups from the respective baseline and that of sham-operated group ($p < 0.05$, Fig 2B). During the period of HFOV, the amplitude

**Table 1. Baseline characteristics of sham-operated and experimental groups at end of stabilization period.**

|  | Sham-operated (n = 6) | HFOV (n = 8) | HFOV+VG (n = 8) |
|---|---|---|---|
| Age (days) | 2.0 (0.6) | 2.3 (0.5) | 1.9 (0.8) |
| Weight (kg) | 1.8 (0.2) | 1.8 (0.1) | 1.9 (0.2) |
| Heart rate (bpm) | 211 (5) | 213 (31) | 226 (26) |
| Mean arterial blood pressure (mmHg) | 66 (4) | 68 (7) | 64 (10) |
| Carotid flow (mL/kg/min) | 35 (8) | 39 (9) | 39 (8) |
| Cerebral oxygenation (%) | 42 (4) | 41 (5) | 38 (7) |
| Cardiac index (mL/kg/min) | 94 (61) | 98 (20) | 96 (27) |
| Stroke volume index (mL/kg) | 0.5 (0.3) | 0.5 (0.1) | 0.4 (0.2) |
| Tau (ms) | 20 (2) | 18 (4) | 18 (2) |
| dP/dt max (mmHg/s) | 3875 (20) | 4078 (18) | 3961 (16) |

HFOV, High frequency oscillatory ventilation; HFOV+VG, High frequency oscillatory ventilation with volume guarantee

Data are presented as mean (standard deviation)

No significant differences (all p>0.05).

was higher in the HFOV+VG group than that of HFOV group (p<0.05), whereas the tidal volume was not different (Fig 3).

## Changes in hemodynamic parameters and cerebral tissue oxygenation

The heart rate of all groups remained largely unchanged throughout the experiment with no significant differences between groups (p>0.05, Table 3). The mean blood pressure of all groups declined gradually over time. Both sham-operated and HFOV+VG groups had significantly lower mean blood pressure than their baseline values after 120 min of the experimental period (63±5 and 65±10 at baseline and 49±4 and 49±10 mmHg at 120 min, respectively; both p<0.05, Table 3). By the end of the experiment, the mean blood pressure of the HFOV+VG group was lower than that of HFOV group (35±12 vs. 52±9 mmHg, p<0.05; Table 3). Right common carotid blood flow remained unchanged in sham-operated and HFOV groups throughout the experimental period (Fig 4A). The common carotid blood flow of the HFOV +VG group decreased over time and was significantly lower than its baseline value after 120 min (20±12 vs. 38±7 ml/kg/min at baseline, p<0.05; Fig 4A). By the end of the experiment, the carotid blood flow of HFOV+VG was significantly lower than the HFOV group (14±10 vs. 30 ±23 ml/kg/min, respectively; p<0.05). Similarly, $CrSO_2$ of the HFOV+VG group declined gradually with time and was significantly lower than that of the HFOV group by the end of the experiment (30±11% vs. 43±10%, respectively; p<0.05; Fig 4B).

Changes in LV function evaluated using a Millar® catheter are summarized in Fig 5. The CI of both experimental groups decreased after lung lavage and remained largely unchanged throughout the recovery period; however, the CI of the HFOV+VG group was lower than that of the sham-operated group at 240 min (88±64 vs. 125±77 ml/kg/min, respectively; p<0.05; Fig 5A). There was no difference in stroke volume, changes of tau, or dP/dt max between intervention groups (Fig 5B–5D). No differences were observed in ejection fraction, stroke work, dP/dt min, end-diastolic volume and pressure obtained from Millar® catheter recordings (S1 File).

## Changes in blood gas analyses

Arterial blood gas parameters (Table 2) demonstrated a significantly lower pH in HFOV and HFOV+VG piglets 60 min after lavage compared to sham-operated piglets (both p<0.05), and

**Table 2. Blood gas changes of sham-operated and experimental groups before and after saline lung-lavage.**

|  | Sham-operated (n = 6) | HFOV (n = 8) | HFOV+VG (n = 8) |
|---|---|---|---|
| **pH** | | | |
| Baseline | 7.41 (0.04) | 7.40 (0.07) | 7.39 (0.07) |
| 0 min | 7.39 (0.07) | 7.35 (0.10) | 7.30 (0.07) |
| 60 min | 7.40 (0.07) | 7.28 (0.10)*# | 7.22 (0.13)*# |
| 240 min | 7.36 (0.07) | 7.28 (0.11)* | 7.20 (0.18)*# |
| **$P_aO_2$ (mmHg)** | | | |
| Baseline | 62 (8) | 63 (8) | 65 (14) |
| 0 min | 61 (3) | 71 (17) | 67 (9) |
| 60 min | 63 (11) | 93 (18) | 110 (58)# |
| 240 min | 62 (11) | 74 (28) | 81 (27) |
| **$P_aCO_2$ (mmHg)** | | | |
| Baseline | 39 (4) | 40 (5) | 42 (5) |
| 0 min | 38 (7) | 42 (9) | 43 (6) |
| 60 min | 35 (10) | 48 (10)*# | 52 (10)*# |
| 240 min | 35 (10) | 44 (5) | 46 (11) |
| **Base excess (mmol/L)** | | | |
| Baseline | -2 (1) | 0 (4) | 0 (4) |
| 0 min | -1 (1) | -2 (3) | -5 (3)* |
| 60 min | -1 (3) | -5 (3)* | -7 (5)* |
| 240 min | -2 (3) | -5 (6)* | -9 (9)*# |
| **Lactate (mmol/L)** | | | |
| Baseline | 4.7 (1.8) | 4.0 (1.2) | 5.1 (1.9) |
| 0 min | 4.1 (1.2) | 4.1 (1.6) | 5.3 (1.5) |
| 60 min | 4.6 (0.6) | 4.1 (1.9) | 5.8 (2.7) |
| 240 min | 3.9 (3.4) | 4.7 (3.4) | 7.6 (4.9) |

HFOV, High frequency oscillatory ventilation; HFOV+VG, High frequency oscillatory ventilation with volume guarantee

Data are presented as mean (standard deviation)

*p<0.05 vs. respective baseline

#p<0.05 vs. sham-operated group.

pH did not improve throughout the recovery period in HFOV and HFOV+VG piglets. The arterial pH of the HFOV+VG group was lower than the sham-operated group at 240 min. Base excess was lower than its own baseline values in both lavaged groups after 60 min of intervention (both p<0.05), and the base excess value of HFOV+VG was lower than sham-operated group at the end of the experiment (p<0.05; Table 2). Plasma lactate levels were not different between HFOV and HFOV+VG groups throughout the experiment (Table 2). Arterial partial pressure of carbon dioxide ($P_aCO_2$) and oxygen ($P_aO_2$) levels did not differ significantly between HFOV groups (p>0.05; Table 2).

## Tissue biochemical assays

Markers of tissue injury were evaluated in samples of brain, right lower lung lobe and left ventricle. Concentrations of IL-8 or TNF-α in lung samples were not significantly different between HFOV and HFOV+VG groups. However, both ventilated groups had significantly higher lung IL-8 concentrations than sham-operated piglets (Fig 6). Tissue lactate

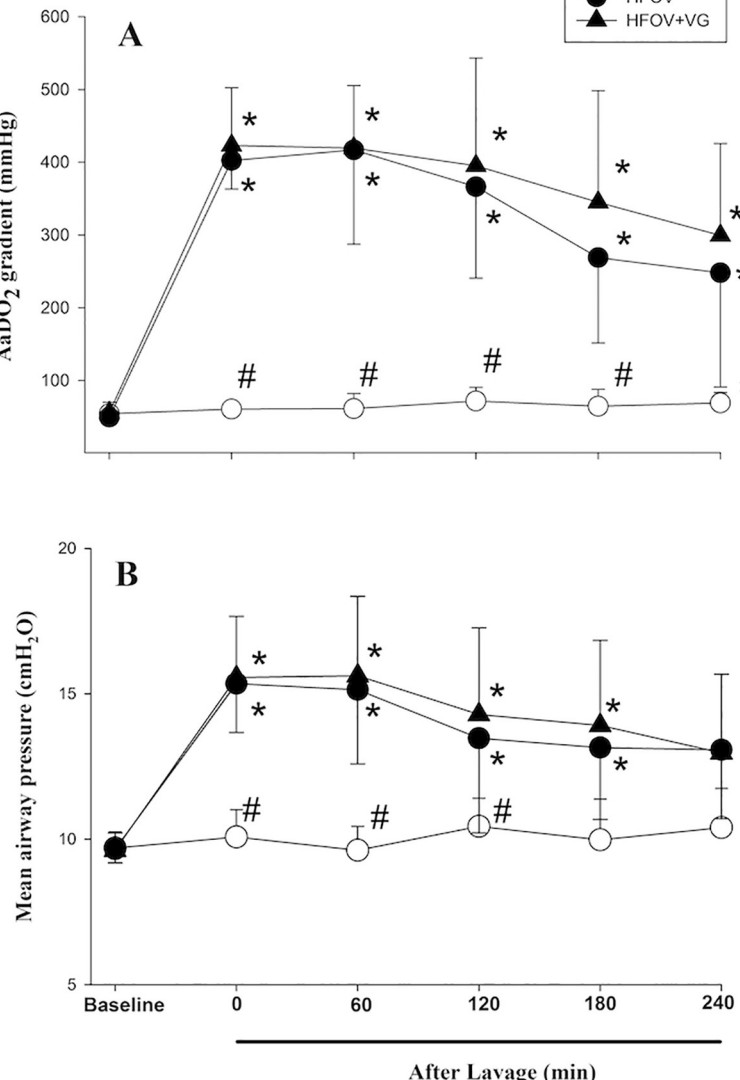

**Fig 2.** Changes in (A) alveolar-arterial oxygen difference ($AaDO_2$) and (B) mean airway pressure. Comparing sham-operated, high frequency oscillatory ventilation (HFOV), and HFOV with volume guarantee (+VG) groups. Data presented as mean with standard deviation error bars. $^*p < 0.05$ vs. respective baseline; $^\#p < 0.05$ vs. both HFOV and HFOV+VG groups.

concentrations and GSSG/GSH ratio in brain and LV tissues were not significantly different among groups (S1 File).

## Discussion

Invasive positive pressure ventilation is commonly used in sick neonates with potential adverse effects on cardiac function. Ventilators have become very accurate and powerful in their ability to ventilate and oxygenate [19]. This study was conducted to assess the hemodynamic impact of HFOV with and without VG, which is relevant to some of the most critically ill neonates who are managed with HFOV.

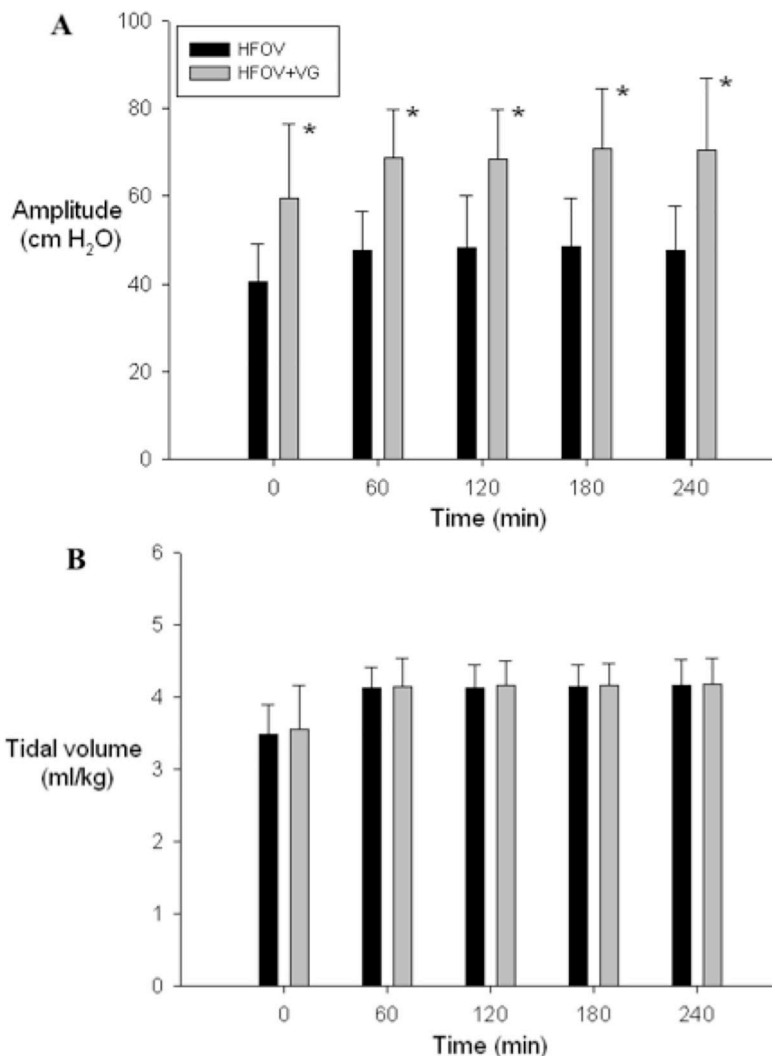

**Fig 3.** Changes in (A) amplitude and (B) tidal volume. Comparison between high frequency oscillatory ventilation (HFOV) and HFOV with volume guarantee (+VG) groups. Data presented as mean with standard deviation error bars. *$p < 0.05$ vs. HFOV group.

Although we expected HFOV+VG to have a more negative impact on cardiac function, there were no significant differences in measures of LV function at any time point between HFOV with and without VG. This included parameters of LV systolic and diastolic function, which are known to be load dependent. Piglets ventilated with HFOV and HFOV+VG had comparable MAP and $AaDO_2$ throughout the experiment suggesting similar positive intrathoracic pressures and severities of lung injury induced by saline lavage. As a consequence, both ventilation approaches likely contributed similarly to changes in preload and afterload, thus resulting in a comparable impact on ventricular function [6].

While we did not observe differences in cardiac function, HFOV+VG had a negative impact on right common carotid blood flow and $CrSO_2$ when compared to HFOV. Several factors are known to influence cerebral perfusion during HFOV. The level of arterial $CO_2$ is known to affect cerebral vasodilation with a directly proportional relationship; hypercapnia increases

**Table 3. Heart rate and mean blood pressure of sham-operated and experimental groups.**

| | Sham-operated (n = 6) | HFOV (n = 8) | HFOV+VG (n = 8) |
|---|---|---|---|
| Heart Rate (bpm) | | | |
| Baseline | 211 (15) | 210 (31) | 226 (26) |
| 0 min | 208 (35) | 205 (30) | 225 (21) |
| 60 min | 213 (30) | 228 (25) | 222 (33) |
| 240 min | 236 (29) | 220 (24) | 215 (39) |
| Mean blood pressure (mmHg) | | | |
| Baseline | 66 (4) | 69 (5) | 65 (10) |
| 0 min | 63 (5) | 69 (14) | 61 (8) |
| 60 min | 55 (10) | 73 (13)[#+] | 56 (11) |
| 240 min | 43 (7) | 52 (9)[+] | 35 (12) |

HFOV, High frequency oscillatory ventilation; HFOV+VG, High frequency oscillatory ventilation with volume guarantee

Baseline = before lung lavage, time points = after lung lavage

Data are presented as mean (standard deviation)

[#]p<0.05 vs. sham-operated group

[+]p<0.05 vs. HFOV+VG group

cerebral blood flow [20, 21]. However, we did not observe a significant difference in $P_aCO_2$ levels between our experimental groups. Hypoxia is also known to increase cerebral blood flow via cerebral vasodilation [22], though our groups demonstrated similar levels of oxygenation and were not hypoxic. Cerebral autoregulation may have played a role in our observations. While cerebral autoregulation is observed in piglets, it is impaired below a mean blood pressure of 40–50 mmHg [21]. Given that the mean blood pressure in HFOV+VG piglets at 240 min was significantly lower than the HFOV group (35 vs. 52 mmHg, respectively), this could result in diminished right carotid artery blood flow which would lead to reduced oxygen delivery culminating in reduced cerebral oxygenation over time (Table 3). We also observed elevated amplitude in the HFOV+VG group throughout the period of HFOV compared with that at the corresponding period in HFOV animals. It is uncertain if the higher amplitude was related to lower cerebral perfusion observed in the HFOV+VG treated piglets. This observation is interesting and should generate further research to understand the relationship between HFOV+VG and cerebral perfusion, which was under-studied.

Systemic blood pressure is a function of cardiac output and systemic vascular resistance [23], and we do not know if the latter was altered in the experimental groups since it was not measured. It is interesting to note that, in addition to significantly lower blood pressure, arterial pH and base excess were lower and plasma lactate was higher at 240 min in the HFOV +VG compared to HFOV group, although the study was underpowered to detect statistical significance in these differences. These findings suggest piglets of the HFOV+VG group were more critically ill at 240 min and, indeed, had lower LV CI over time compared to its baseline. We speculate that the fluctuating amplitude during HFOV+VG may result in fluctuating positive intrathoracic pressure. However, Kamitsuka et al demonstrated insignificant alveolar pressure changes with changes in amplitude in HFOV-treated rabbits [24]. The relationship between LV contractile function, amplitude and MAP pressure is complicated and needs further investigation. Since MAP remains the strong effector of cardiac output, we observed no effect of HFOV+VG on the LV CI and cardiac parameters measured by Millar® catheter when compared with HFOV. Further, these pressure fluctuations may only have been significant if

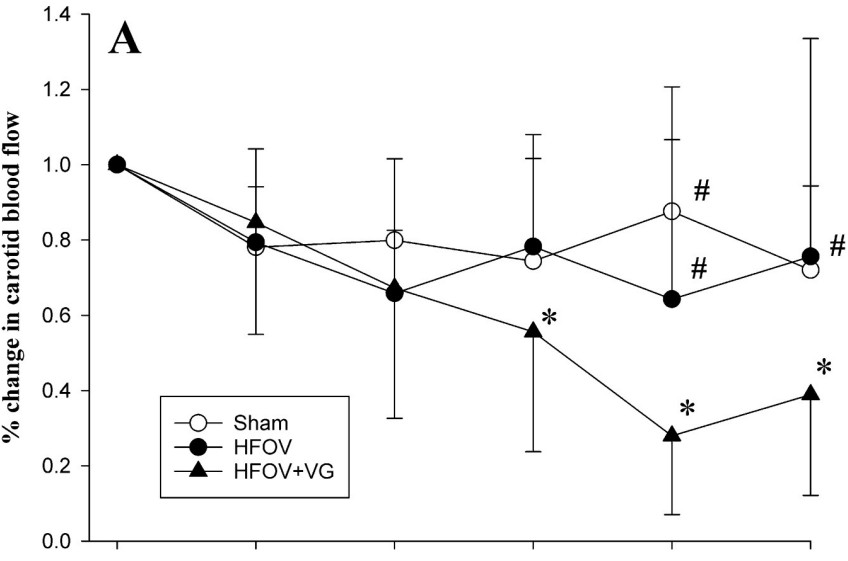

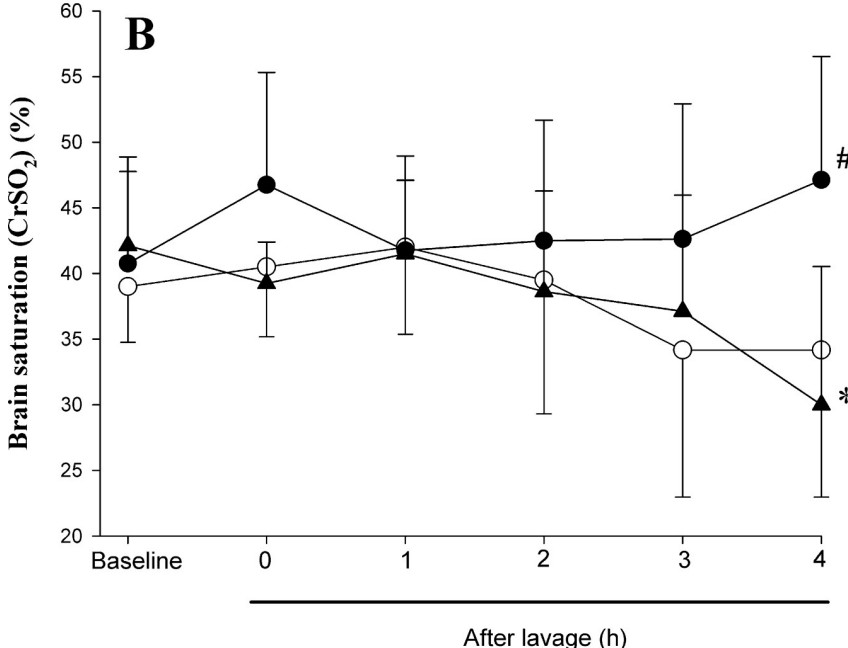

**Fig 4.** Changes in (A) right common carotid blood flow index and (B) brain oxygen saturation (near-infrared spectroscopy). Comparing sham-operated, high frequency oscillatory ventilation (HFOV), and HFOV with volume guarantee (+VG) groups. Data presented as mean with standard deviation error bars. *$p < 0.05$ vs. respective baseline; #$p < 0.05$ vs. HFOV+VG group. $CrSO_2$ = infrared spectroscopy of cerebral oxygen saturation.

respiratory mechanics such as compliance and/or resistance differed between piglets, as found in preterm lambs by Pillow et al [25].

Our findings of differences in carotid blood flow and $CrSO_2$ are especially interesting in the clinical care of the neonate. Markers of regional perfusion, such as urine output, often suggest the presence of reduced blood flow and/or oxygen delivery before more direct measures of

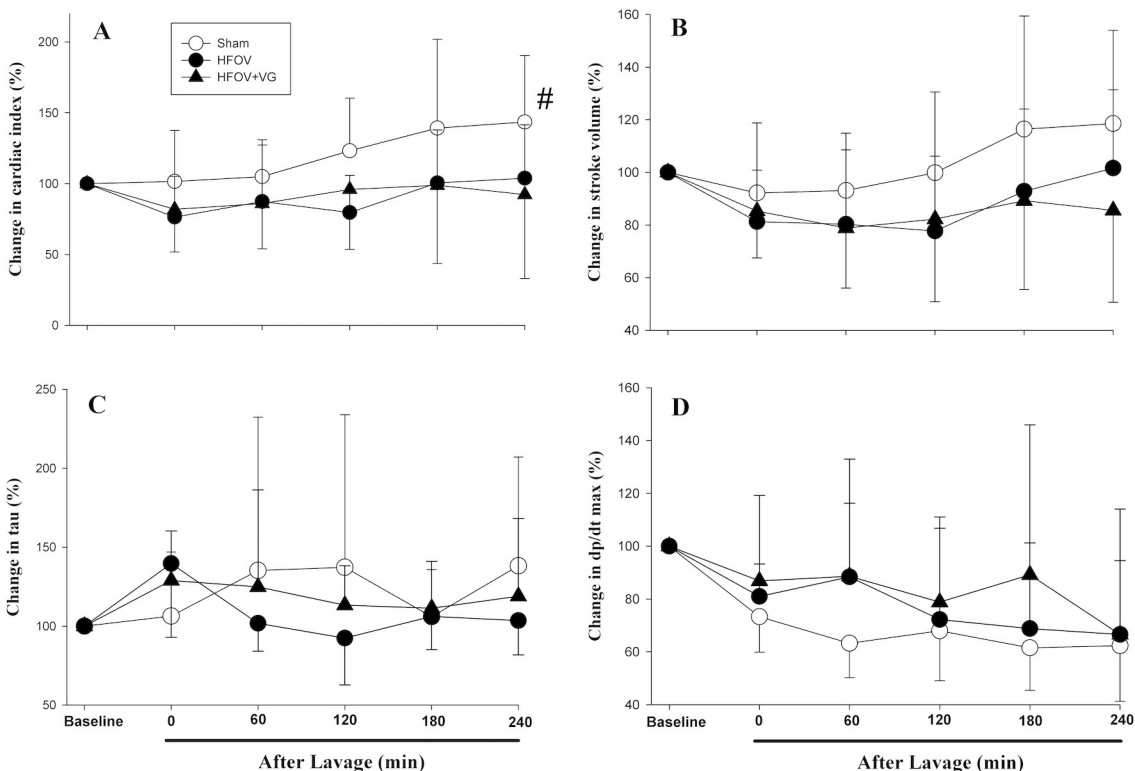

**Fig 5.** Changes in (A) cardiac index, (B) stroke volume, (C) tau, and (D) dP/dt max. Comparing sham-operated, high frequency oscillatory ventilation (HFOV), and HFOV with volume guarantee (+VG) groups. Data presented as mean with standard deviation error bars. #p<0.05 vs. both HFOV and HFOV+VG groups.

ventricular function [26]. In our experiment, regional markers revealed evidence of compromise before the systemic markers (pH, base excess, lactate) and LV CI, which may have become more apparent over a longer experimentation time or with a larger sample size. We are not certain if the lower blood pressure and cerebral perfusion might affect inotropic use and neurological outcomes, respectively, in HFOV+VG treatment. Nevertheless, our findings suggest potential differences between ventilation modes that justify further exploration in

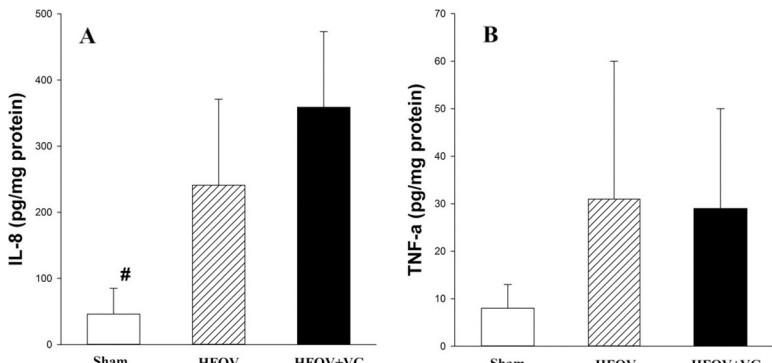

**Fig 6.** Bar plot showing immediate post-mortem lung tissue analysis for (A) IL-8 and (B) TNF-α. Comparing sham-operated, high frequency oscillatory ventilation (HFOV), and HFOV with volume guarantee (+VG) groups. Data presented as mean with standard deviation error bars. #p<0.05 vs. both HFOV and HFOV+VG groups.

either further animal studies or novel human studies with non-invasive monitoring such as near-infrared spectroscopy and echocardiography.

To our knowledge, no study has examined the hemodynamic effects of HFOV with and without VG. Studies on systemic and/or regional hemodynamic effects of HFOV when compared to CMV have suggested no difference in cardiac/systemic and regional effects in adult rabbits with induced surfactant deficiency at similar MAPs [27], no difference in cardiac/systemic and regional effects in baboons with similar MAPs [28], and reduced cardiac function in neonates with a higher MAP on HFOV [7].

Recent studies comparing HFOV with and without VG have focused on stability of ventilation and oxygenation measures, notably $P_aCO_2$ and $S_pO_2$. So far these have been pilot studies in neonates that run over a period of hours. Though not the primary focus of our study, we found no significant differences in $P_aO_2$ and $P_aCO_2$ levels between HFOV groups (Table 2). Enomoto et al reported reduced fluctuation (based on the standard deviation) in $SpO_2$, minute ventilation, and $DCO_2$ (calculated marker of alveolar ventilation) with HFOV+VG when compared to HFOV in six preterm infants with birthweight <1000 grams and postnatal age >28 days ventilated for 6 hours on each mode [12]. In a randomized cross-over study, Iscan et al included 20 preterm infants <32 weeks' gestation and <6 hours of age after surfactant administration for RDS, and reported that HFOV+VG resulted in higher mean tidal volume and $DCO_2$, more consistent tidal volume, and less fluctuation of $P_aCO_2$ compared to HFOV, with similar $FiO_2$ [13]. Though our study did not measure stability of oxygenation and ventilatory measures, there was no significant difference between mean $P_aCO_2$ or $P_aO_2$ levels when comparing HFOV and HFOV+VG modes.

There are several limitations to the present study.

## Confounders and applications of the animal model

As it is an animal study of term piglets aged 1–3 days with a 4-hour experimental time frame, the extrapolation is limited to human late preterm or early term infants with short-term ventilation. Hemodynamic differences between HFOV modes may become more evident over a longer experimentation time, as may be indicated by the decrease in mean blood pressure and $CrSO_2$. Anesthetic medications might depress the cardiovascular function including decreased blood pressure [29, 30]. We did not record the total anesthetic use in each piglet and therefore could not assess the confounding effects of anesthesia. Although the saline lung lavage model has been validated in terms of lung injury and $AaDO_2$ gradient achieved, there was a gradual decrease in $AaDO_2$ gradient over time up to 150–200 mmHg by 240 min. This suggests a transient lung injury model including pulmonary edema secondary to retained saline from the lavage, in addition to surfactant deficiency, that improves over the short time frame of experimentation. We also did not provide exogenous surfactant (the standard treatment for RDS) or inotropes, which are not uncommonly used and have confounding hemodynamic effects.

## Lung recruitment maneuver in HFOV

A recruitment maneuver was not performed when starting HFOV since it was decided that it would destabilize the already critically ill piglets by increasing the transpulmonary pressure gradient. Further, there were a few other factors that limited the inclusion of lung recruitment maneuvers. These factors included the increasing saline lavage duration, lengthy procedure, and interactive compromise of the cardiovascular system which might not tolerate the incorporation of a recruitment maneuver. Recruitment may have confounding effects including a temporary increase in transpulmonary pressure, ventilator parameters (lowered MAP, amplitude and tidal volume), and $pO_2$ and $pCO_2$, any of which might affect the systemic and

regional hemodynamics. Lung recruitment could be incorporated into future studies with a modified animal model.

### Biventricular function

Cardiac function data was limited to the LV parameters. Right ventricular function, intra- and extracardiac shunts, and valvular regurgitation (notably aortic regurgitation related to the intraventricular catheter), and the intraventricular catheter position could have been assessed non-invasively with echocardiography. Given the interdependency of the right ventricle and LV, including the LV's preload dependence on right ventricular output, information on right ventricular function is integral to explaining LV function [6].

### Other factors

Flow measurements of different cerebral arteries may have provided more information regarding specific regional cerebral blood flow. This is notable considering the Millar® catheter may theoretically obstruct blood flow through the left common carotid artery to an extent, though this was not demonstrated in any studies reviewed, and would have not explained differences between ventilation modes. The tissue biomarkers of ischemia, injury and inflammation represent the cumulative impact during the experimental period, not the effects at different time points. Ventilation parameters were standardized in terms of weaning, and the MAP was kept consistent between modes given it could be a confounding variable in terms of hemodynamic effects. This may limit the extrapolation to clinical use. Nevertheless, $P_aCO_2$ levels were similar between HFOV and HFOV+VG modes. The ventilator used was the Fabian HFO (Acutronic, Hirzel, Switzerland) which uses a membrane to generate oscillations, compared to the more commonly studied Babylog VN500 which uses a Servo valve to create oscillations via pulsed inspiratory and expiratory flow [31]. This may limit generalizability based on differing mechanisms of oscillation, though this would need to be studied.

In our newborn term piglet model of moderate to severe RDS, there was no effect of HFOV or HFOV+VG on direct measures of cardiac function. However, there was a significant negative effect of HFOV+VG on mean blood pressure over time, carotid blood flow, and cerebral oxygenation. Further research is required to confirm these results and understand potential differing effects of ventilatory modes on cerebral hemodynamics.

## Supporting information

**S1 Checklist. ARRIVE guidelines checklist.**
(PDF)

**S1 Table. Spreadsheet data from HFOV group.**
(PDF)

**S2 Table. Spreadsheet data from HFOV+VG group.**
(PDF)

**S3 Table. Spreadsheet data from sham-operated group.**
(PDF)

**S1 File. Tabulated data of mean ± standard deviation of Millar® data, tissue lactate, and tissue GSSG/GSH ratio.**
(PDF)

## Acknowledgments

The authors would like to thank Min Lu, BSc, for his invaluable contribution to the research study. Min was involved in the surgical procedures and assisted with multiple aspects of the experiment including biochemical tissue analyses.

## Author Contributions

**Conceptualization:** Jagmeet Bhogal, Anne Lee Solevåg, Tze-Fun Lee, Georg M. Schmölzer, Po-Yin Cheung.

**Data curation:** Jagmeet Bhogal, Megan O'Reilly, Tze-Fun Lee.

**Formal analysis:** Jagmeet Bhogal, Tze-Fun Lee, Po-Yin Cheung.

**Funding acquisition:** Po-Yin Cheung.

**Investigation:** Jagmeet Bhogal, Megan O'Reilly, Po-Yin Cheung.

**Methodology:** Jagmeet Bhogal, Megan O'Reilly, Tze-Fun Lee, Georg M. Schmölzer, Po-Yin Cheung.

**Project administration:** Jagmeet Bhogal, Anne Lee Solevåg, Tze-Fun Lee, Georg M. Schmölzer, Po-Yin Cheung.

**Resources:** Megan O'Reilly, Tze-Fun Lee, Po-Yin Cheung.

**Software:** Jagmeet Bhogal, Tze-Fun Lee.

**Supervision:** Jagmeet Bhogal, Tze-Fun Lee, Chloe Joynt, Lisa K. Hornberger, Georg M. Schmölzer, Po-Yin Cheung.

**Validation:** Jagmeet Bhogal, Megan O'Reilly, Tze-Fun Lee, Po-Yin Cheung.

**Visualization:** Jagmeet Bhogal, Tze-Fun Lee.

**Writing – original draft:** Jagmeet Bhogal, Po-Yin Cheung.

**Writing – review & editing:** Jagmeet Bhogal, Anne Lee Solevåg, Megan O'Reilly, Tze-Fun Lee, Chloe Joynt, Lisa K. Hornberger, Georg M. Schmölzer, Po-Yin Cheung.

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
