## [Decision Letter · Decision Letter 0]

8 Sep 2020

PONE-D-20-20869

Hemodynamic effects of high frequency oscillatory ventilation with volume guarantee in a piglet model of respiratory distress syndrome

PLOS ONE

Dear Dr. Bhogal,

Thank you for submitting your manuscript to PLOS ONE. After careful consideration, we feel that it has merit but does not fully meet PLOS ONE’s publication criteria as it currently stands. Therefore, we invite you to submit a revised version of the manuscript that addresses the points raised during the review process.

As noted by the reviewers, there are several issues that need to be addressed. Both question why a recruitment maneuver was not performed. In addition, reviewer #1notes that the main factor influencing cardiac output is the transmural pressure, not MAP. Please discuss this comment. There are other areas that I identified that also need to be addressed. Why is the BP of the SHAM animals so low at 240 min? Since cardiac output was unchanged, the level of anesthesia could explain the decrease in cerebral NIRS and carotid blood flow. This needs to be discussed. How was anesthesia adjusted? Was it the same for all animals? In Figure 3B, the y-axis is labeled change in cerebral NIRS but the text refers to absolute values. Please graph the absolute values.

We look forward to receiving your revised manuscript.

Kind regards,

Richard Bruce Mink

Academic Editor

PLOS ONE

Journal Requirements:

Reviewers' comments:

Reviewer's Responses to Questions

**Comments to the Author**

1. Is the manuscript technically sound, and do the data support the conclusions?

Reviewer #1: No

Reviewer #2: Yes

2. Has the statistical analysis been performed appropriately and rigorously? 

Reviewer #1: Yes

Reviewer #2: Yes

3. Have the authors made all data underlying the findings in their manuscript fully available?

Reviewer #1: No

Reviewer #2: Yes

4. Is the manuscript presented in an intelligible fashion and written in standard English?

Reviewer #1: Yes

Reviewer #2: Yes

5. Review Comments to the Author

Reviewer #1: In the hypothesis the authors established that during HFOV combined to VG, amplitude pressure is titrated to maintain tidal volume and this can affect cardiac output. The authors speculate that the only finding of a decrease in the Carotid artery flow index and NIRS in the HFOV + VG group is due to speculate fluctuating of amplitude during HFOV+VG may resulting in fluctuating positive intrathoracic pressure, which would contribute to changing preload for the right ventricle that ultimately impacts the LV preload. LV afterload may similarly fluctuate in this context resulting overall in a negative impact on cardiac output. But the authors didn’t find any differences between the intervention groups in the stroke volume, changes of tau, dP/dt max, ejection fraction, stroke work, dP/dt min, end-diastolic volume and pressure obtained from Millar® catheter .Also it has been demonstrated in an experimental model that delta P generated by the ventilator is not transmitted into the airways, also as the authors demonstrated the is no different effect over the time on CO when compared Vg vs no VG.

Animal model, it is missed a maneuver of lung recruitment after BAL.

Method, the measurement of delta pressure is missed, this is the main variable during HFOV with the VG modality, if VT is similar between both modalities, then delta P should be also similar and no differences at all will be found, unless the ventilator used clearly work different during the VG modality. It is not well described the ventilator used for the study, and how the VG works, if this is the Fabian device, not many studies have been done with this device and as it is well known, the device used during HFOV clearly can affect the results. Most of the studies published to date used the VN500 from Dragger, so results can’t be clearly compared.

Results, in additional information from the animals, the two animals with the lower NIRS values in the Vg group are the two who had lower CO at the stable condition. CA data from the VG group are missed in two animals at 240 min and very low in three at 120 min, so comparison is quite difficult to be done against the non VG group. Minute volume is represented during the HFOV but this value is of no interest during this modality of ventilation as the ventilation is more accurately represented by the DCO2.

pH values in the VG group showed a very acidotic situation in one animal from the beginning of the study.

Final comments, this is an interest study exploring the potential different effect of the VG modality when used in combination with HFOV instead to not using it. The main problem is related to the design as if the classical variable during standard HFOV is related to the Vt and during VG it is related to delta P, if the ventilator, tubbing and lung condition are similar and there is no interaction between the animal and the ventilator, then both modalities are exactly the same with minimal fluctuation in delta P to maintain the set Vt in the VG added to HFOV, and minimal fluctuation in Vt when delta P is set in the no VG modality.

So the expected differences between both would be mostly related to changes in the lung and patient condition during ventilation. If the patient’s lung is stable, then both modalities will work equal at similar Vt.

Reviewer #2: The presented study has examined the hemodynamic effects of HFOV with and without VG in the newborn piglets with RDS. In general, the study is well established and written .

-Open lung strategy is the key element of lung protective strategies. Generally, in cases with diffuse atelectasis, as in the RDS, lung recruitment maneuver at the initiation of HFOV or HFOV-VG stabilise alveoli and improves oxygenation. Besides a few exceptions, current practice at the initiation of HFOV or HFOV-VG involves lung recruitment maneuvers. Why the authors did not optimise lung volumes using lung recruitment maneuver ? ... If open lung strategy had been used , piglets posssibly would need lower MAP and and may need lower amplitude / TVhf levels to improve oxygen delivery and to achieve normocarbic levels. On the other side, recruitment could have temporarily increased transpulmonary pressure and could have effected the results. The effect of open lung strategy may be dependent of the selected HFOV strategy (With or without V).

It would be good to discuss the possible effects of the lung recruitment maneuvers to the results.

Besides lung recruitment, results can not directly be extrapolated to newborn babies with RDS without surfactant administration and use of inotropic drug.

- MAP is the strongest effector of intra thorasic pressure, however TVhf do not directly effects it. Therefore, systemic circulation is possibly efffected by MAP levels rather than VThf levels. Increased hypotension and lower perfusion findings with HFOV-VG mode compared to HFOV mode ( using similar MAP levels ) sounds interesting and needs to be emphasized. Is there a clear relationship between amplitude and intrathorasic pressure since intrathorasic is mainly effected by MAP

It is valuable in terms of contribution to the literature, but some issues need to be explained.

6. PLOS authors have the option to publish the peer review history of their article (what does this mean?). If published, this will include your full peer review and any attached files.

Reviewer #1: No

Reviewer #2: No

---

## [Author Response · Author response to Decision Letter 0]

27 Jan 2021

Responses to Reviewers’ and Academic Editor’s Comments

We are grateful to the insightful comments of both reviewers and the academic editor. The reviewers also complimented our interesting observations and their contribution to the literature. We believe that we have adequately addressed these comments in the revised manuscript and improved the scientific merit of the submission, which becomes a technically sound piece of scientific research with data that supports the conclusions.

Specific responses to each comment:

Reviewer #1: 

1. HFOV+VG vs. HFOV and carotid artery flow index and NIRS: Thank you for the comment and we agree that our speculation that the reason for the change in carotid blood flow was related to the fluctuating amplitude during HFOV+VG was weak as the reviewer cited that delta P generated by the ventilator was not transmitted into the airways in an experimental model (Kamitsuka et al 1990). We included this in our discussion on page 19 (line 374-378). We also admitted that there was no effect of HFOV+VG on the CO and cardiac parameters measured by Millar catheter when compared with HFOV. However, we did not study the mechanistic pathway between fluctuating amplitude and cerebral perfusion in the absence of any significant effect on the intrathoracic pressure, CO and other cardiac parameters. 

2. Amplitude and Tidal Volume: As the point regarding the amplitude and tidal volume was commented by both reviewers, we further examined the changes in amplitude and tidal volume during HFOV+VG and HFOV modes in these animals after saline lavage. We found that the amplitude was higher in the HFOV+VG group than that of the HFOV group whereas the tidal volume was not different between groups. We are not certain about the relevance of this observation. The results (P.13, line 240-242 and 250-253, Fig.3) and discussion (P.19 line 360-365) had been added in the revised manuscript. This point is important and should generate further research to understand the relationship between HFOV+VG and cerebral perfusion, which was under-studied. 

3. Animal model: The lack of lung recruitment maneuver after saline lavage is a limitation of our study although this clinical practice might further destabilize our critically sick animals. We have included this in the methods (P.7, line 146-147) and discussion on page 22 (line 428-437). 

4. Oscillatory mechanism: The discussion of oscillatory mechanisms of the Fabian HFO vs. Babylog VN500 was also included on P.23 line 453-457. We agree that this is a main variable that may be different during HFOV with the VG mode. 

5. Results: Thanks to this reviewer who pointed out the 1-2 outliers in the groups regarding the NIRS, CO, carotid blood flow and arterial pH. Although the increased variance precluded us from identifying significant differences between groups, passing the normal distribution and equal variance pre-priori testing indicated the appropriate use of statistical analyses. The reviewer also graciously explained that DCO2 is superior to MV as a measure of ventilation in HFOV, so we have added the DCO2 data to the supplemental files immediately below the MV data (S1_Table, S2_Table). 

6. Final comments: We are grateful to this reviewer who commented that this is an interesting study exploring the potential different effect of the VG mode when used in combination with HFOV instead of not using it. S/he highlighted the key concept that changes in the lung and patient condition during ventilation may result in fluctuation in delta P to maintain the set tidal volume in the VG mode and therefore may account for the potential differences in the observation of hemodynamics. If the patient’s lung is stable, then both modes will work equally at similar tidal volumes. This important point has been highlighted in the revised manuscript on page 19 line 379-381.

Reviewer #2: 

1. Open lung strategy is the key element of lung protective strategies: We are thankful for this insightful comment. As indicated by reviewer #1, this is a limitation regarding the translational value of this animal model. We adopted the model with modifications in newborn piglets. There were a few factors that limited the inclusion of lung recruitment maneuvers. These factors included the increasing saline lavage duration, lengthy procedure and interactive compromise of the cardiovascular system which might not tolerate recruitment maneuvers. Further improvements in the model will be needed. This limitation has been added to the revised version on page 22 (line 428-437). We also discussed the confounding effects of lung recruitment maneuvers on temporarily increased transpulmonary pressure, ventilator parameters including lowered MAP, amplitude and tidal volume, and pO2 and pCO2, which might affect the systemic and regional hemodynamics. The interesting speculation of effects of open lung strategy depending on HFOV mode was also included.

2. Extrapolation of results to newborn babies with RDS: We thank this reviewer who pointed out the limitation that clinically newborn babies with RDS have received surfactant and usually require the use of inotropic agents. This limitation has been added to the discussion on page 21-22 line 425-427.

3. Relationship between HFOV on systemic hemodynamics: We agree with this reviewer that MAP, not tidal volume/delta P, is the strongest effector of intrathoracic pressure (P.19, line 376-378). This supports our observation that the CO was not affected by the mode of HFOV as both groups had similar MAP. As this reviewer indicated, the increased hypotension and lower perfusion findings with HFOV-VG mode compared to HFOV mode (using similar MAP levels) are interesting and require further investigation. The relationship between amplitude and intrathoracic pressure and MAP are interesting and need further investigation if this is the etiology for worsened hemodynamics (P.19, line 375-376).

Academic Editor:

1. Lung recruitment: This has been added in the methods (P.7, line 146-147) and discussion (P.22, line 428-437).

2. The relationship between cardiac output, transmural pressure, delta P (amplitude) and mean airway pressure: As mentioned above that both reviewers had different opinions on the relationship, we have added the complex relationship and the additional observations on the amplitude and tidal volume during HFOV+VG and HFOV in the discussion (P.19, line 372-381). 

3. Sham blood pressure low at 240min: Table 3 shows the mean blood pressure of the sham-operated piglets at 240min was 43mmHg, which was lower than baseline but not significantly different between the experimental groups. This could be related to the effect of anesthetic and surgical interventions in this model.

4. Anesthesia: The editor’s comment that anesthesia could be a factor relating to systemic and regional measurements is an excellent one. Although similar anesthetic/analgesic dose infusion ranges were used for each piglet we did not record the precise amounts given so are unable to rule it out as a factor. This has been added to the limitations on page 21, line 418-421. The inclusion of sham-operated piglets could help interpret the hemodynamic effects of two HFOV modes. Nevertheless, all animals were closely monitored throughout the experiments with adjustment of anesthetic and analgesic medications according to their neurological status. This detail has been added to the methods on page 6, line 105-106.

5. CrSO2 graph Figure 3B: We thank the editor for the observation regarding the discrepancy between absolute values in the table and percent change in the figure. The graph has been changed accordingly. 

6. Data not shown: An additional supplemental figure has been added (S4_Table) which contains the previously mentioned ‘data not shown’ in mean ± standard deviation format, which includes Millar® catheter and tissue data.

Sincerely,

Jag Bhogal and Po-Yin Cheung

---

## [Editor Report · Decision Letter 1]

1 Feb 2021

Hemodynamic effects of high frequency oscillatory ventilation with volume guarantee in a piglet model of respiratory distress syndrome

PONE-D-20-20869R1

Dear Dr. Bhogal,

We’re pleased to inform you that your manuscript has been judged scientifically suitable for publication and will be formally accepted for publication once it meets all outstanding technical requirements.

Kind regards,

Richard Bruce Mink

Academic Editor

PLOS ONE
---

## [Editor Report · Acceptance letter]

4 Feb 2021

PONE-D-20-20869R1 

Hemodynamic effects of high frequency oscillatory ventilation with volume guarantee in a piglet model of respiratory distress syndrome 

Dear Dr. Bhogal:

I'm pleased to inform you that your manuscript has been deemed suitable for publication in PLOS ONE. Congratulations! Your manuscript is now with our production department. 

Kind regards, 

on behalf of

Dr. Richard Bruce Mink 

Academic Editor

PLOS ONE